# Direct Numerical Modeling as a Tool for Optical Coherence Tomography Development: SNR (Sensitivity) and Lateral Resolution Test Target Interpretation

**Samuel Lawman** [1,2,*] **and Yao-Chun Shen** [1]

1 Department of Electrical Engineering and Electronics, University of Liverpool, Liverpool L69 3GJ, UK; y.c.shen@liverpool.ac.uk
2 Department of Eye and Vision Science, University of Liverpool, Liverpool L7 8TX, UK
* Correspondence: samlawman@hotmail.com

**Abstract:** Optical Coherence Tomography (OCT) is a growing family of biophotonic imaging techniques, but in the literature there is a lack of easy-to-use tools to universally directly evaluate a device's theoretical performance for a given metric. Modern computing tools mean that direct numerical modeling can, from first principles, simulate the performance metrics of a specific device directly without relying on analytical approximations and/or complexities. Here, we present two different direct numerical models, along with the example MATLAB code for the reader to adapt to their own systems. The first model is of photo-electron shot noise at the detector, the primary noise source for OCT. We use this firstly to evaluate the amount of additional noise present (1.5 dB) for an experimental setup. Secondly, we demonstrate how to use it to precisely quantify the expected shot noise SNR limit difference between time-domain and Fourier-domain OCT systems in a given hypothetical experiment. The second model is used to demonstrate how USAF 1951 test chart images should be interpreted for a given lateral PSF shape. Direct numerical modeling is an easy and powerful basic tool for researchers and developers, the wider use of which may improve the rigor of the OCT literature.

**Keywords:** Optical Coherence Tomography; Optical Coherence Microscopy; numerical modeling; noise; sensitivity; line field; spectral domain; point spread function; resolution; USAF 1951

## 1. Introduction

Optical Coherence Tomography (OCT) [1–3] encompasses a broad family of biophotonic imaging techniques that use Low-Coherence Interferometry (LCI) for axial ranging. Different versions of the technique have been adopted for a wide range of biophotonic applications, including retinal imaging [4], corneal imaging [5], dermatology [6], various endoscope applications [7,8], dentistry [9], angiography [10], and optical coherence elastography (OCE) [11] (and OCE-related techniques [12,13]). It has also found a range of uses outside biophotonics [14–17]. Though the technique is now over 30 years old [18], it is still being improved and developed for new applications, particularly with component technological performance advances making additional measurement concepts realizable in a real-world environment. One such concept, which we will use as an available example here, is Spectral-Domain (SD) Optical Coherence Microscopy (OCM) [19,20], which is essentially an SD-OCT system where microscope objectives are used to obtain higher lateral resolution at the focal plane at the cost of a highly limited depth of field of that focal plane. As the technology of the hardware components improves and their cost is reduced, this technique may become an economically viable direct alternative (with additional 3D visualization benefits) to existing clinical In Vivo Confocal Microscopy (IVCM) systems, which have applications in identifying pathogens in infectious keratitis [21] and screening for Diabetic Peripheral Neuropathy (DPN) [22].

For the development of new OCT devices, tools to calculate the theoretical performance of a system are useful for both the initial design stage and the evaluation of constructed hardware. The vast majority of the existing OCT literature relies (at least partly) on analytical derivations to provide theoretical comparative values. A couple of examples outside the direct scope of this paper are provided in the following. Veselka et al. [23] recently used extensive analytical derivations to model the amplitude of an OCT signal as a function of surface tilt. Monte Carlo simulations of signals from scattering samples are based on some analytical derivations [24]. Within the scope of this paper, the two most highly cited papers [25,26] on theoretical signal-to-noise performance in OCT rely on analytical approaches to estimate the statistical behavior of noise; these two papers are discussed further in the discussion. Usually, analytical approaches depend on specific assumptions, e.g., a Gaussian spectral shape (with $\Delta\lambda << \lambda$) in the commonly quoted "definition" of axial resolution. (The better universal numerical approach is just to take the fast Fourier transform of the effective spectrum). Therefore, exact predictions for a specific device are generally not practical with an analytical approach. Instead, here, we show with two examples that direct numerical modeling can provide a direct theoretical expectation for an exact system. These models are based just on basic/fundamental optical laws. Modern computational tools mean that these are straightforward for the majority of researchers to implement. For such models, any arbitrary set of parameters (e.g., any arbitrary spectral shape or system point spread function) can be inputted to give exact expected experimental outcomes. With this paper, we provide the novel code used for both examples for OCT (and other biophotonic) researchers to adapt to their own needs.

### 1.1. SNR and Sensitivity

In the vast majority of cases, the image Signal-to-Noise Ratio (SNR) (the ratio between the largest signal before saturation and the standard deviation of noise) and absolute sensitivity (the ratio between a hypothetical perfect reflector and the standard deviation of noise (with no sample present, i.e., the smallest absolute reflection detectable)) are key performance metrics of OCT devices. The term SNR is often used loosely in the OCT literature, including commonly to refer to absolute sensitivity. Strictly, the noise in the SNR should be measured when the signal is present. However, for convenience in converting between the SNR and sensitivity, we will use a modified definition with the noise measured without any sample signal present.

The standard approach [25,26] to estimating the theoretical SNR and sensitivity is analytically starting at the light source.

Here, instead, we can simplify it into two simple problems; the noise at the point of detection can be numerically modeled from first principles directly and the optical energy efficiency of all optical components (up to and including the quantum efficiency of the detector). We can do this because, as far as the technology is concerned, any photon not detected (i.e., lost within the system) (excluding due to relative destructive inference in LCI) may as well have not existed, as it provides no information. Likewise, in the analytical equations in the standard approach [25,26], a large proportion of the mathematical detail is the calculation of the photo-electrons detected at the detector. In practice, the transmission efficiency of photons through each part of the system can be directly measured (e.g., with a power meter). Here, we provide a method for directly modeling the expected OCT SNR (and sensitivity) from the (ideally averaged) measured or predicted raw signal.

In addition to its use as a practical system analysis tool, direct numerical modeling can be used to provide detailed insight into more fundamental questions about a technology. It has been established analytically that Fourier domain (FD) (including SD and swept-source systems) is superior to time domain (TD) for SNR (therefore, absolute sensitivity) performance [25,26]. For shot-noise-limited setups, it is predicted that the SNR improvement of FD over TD should be dependent on N (the number of detection points). For any given hypothetical experiment, direct numerical modeling can predict the outcome.

*1.2. USAF 1951 Target*

For OCM, the key system performance metric is the lateral resolution. The gold-standard method for characterizing the lateral resolution of any imaging system is the full determination of the (lateral) Point Spread Function (PSF) (or its Fourier transform, the Modulation Transfer Function (MTF)) [27]. However, especially in the OCT field, it is more convenient to be able to quote a single (per dimension) resolution value, equivalent to the PSF Full-Width at Half Maximum (FWHM), which is standard to quote for the axial resolution. To do this, the USAF 1951 lateral resolution targets are commonly used, with the lowest resolved 3 bar Ronchi grating element indicating a corresponding FWHM value. Direct numerical modeling can be used to show the likely FWHM interpretation for the smallest resolved element.

The United States Department of Defense standard MIL-STD-150A, revised in 1959 [28], defines the "USAF 1951" lateral resolution target that is widely used today. In MIL-STD-150A, Section 3.6.2 defines resolving power as lines per mm (pairs per mm if including negative space as a "negative" line), and the definition of resolved is being able to count the correct number of lines. Section 5.1.1.7 and Figure 7 **of the standard** define the USAF 1951 target, noting that the key parameter referred to is the line (pairs) per mm or line-pair width. The standard does not imply that the single-line width, which is half of the line-pair width, is equivalent to the resolution.

There are a large number of OCT papers that quote single-line width as the lateral resolution (we will not cite these here) that can be found, while there are a few that quote the line pair width [29,30]. Here, using direct numerical modeling of imaging PSF, we will show that the line-pair width is the better (equivalent to the point object separation resolution, given a PSF approximating a Gaussian) method of interpretation.

## 2. Materials and Methods

*2.1. SNR Model*

Figure 1 gives an overview of the noise modeling process. At the detector, the detected signal(s) (average measurement or modeled) can be converted into photo-electrons. With no pre-amplification (if pre-amplification is used, the specified electron well depth needs to be attenuated accordingly) of the signal before Analog-to-Digital Conversion (ADC), a pixel of a camera will be expected to use its full specified electron well depth, $W_e$. Also, in this case, there is no zero-read offset. Therefore, the ADC output, $N_c$, can be converted into photo-electrons with

$$N_e = \frac{W_e}{S} N_c$$

where S is the ADC saturation count (i.e., $2^B$, where B is the bit depth). With an ideal signal converted into photo-electrons, shot noise can be directly modeled by applying a random number generator with a numerical Poisson distribution; here, we used MATLAB's poissrnd function. This noise generation works, for each simulated raw data point, by inputting the seed average expected number of photo-electrons (generally not an integer). The Poisson random number generator then returns a random integer value with a Poisson probability distribution for that seed, mimicking shot noise directly.

Here, we did not add additional noise sources, such as spectral Random-Intensity Noise (RIN) (inherent to most current super-continuum light sources) [31] and camera electronic read noise, into the model. Instead, we used the modeled shot noise value to experimentally quantify the effect of other noise in the system. Alternatively, if the statistics of these noises are adequately described, both could also be added into the numerical simulation.

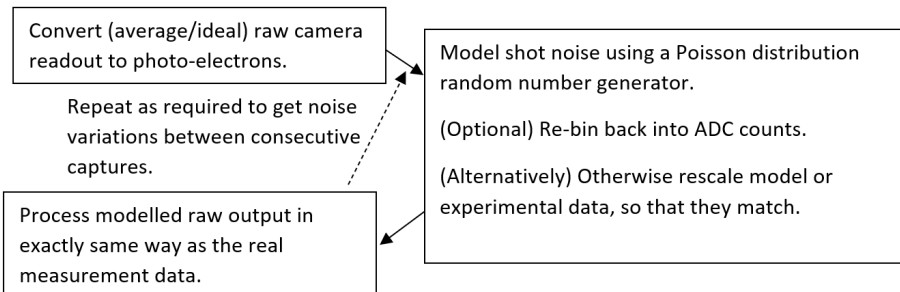

**Figure 1.** Flowchart of the direct numerical modeling process for OCT noise.

After that, the modeled photo-electron signal can be re-binned into the detector bit depth, though for most cases, the pseudo-noise due to the bit depth will be negligible compared to the shot noise. The modeled raw signals are then processed in the same way as the real signal.

For the OCT SNR, the numerical modeling follows the measurement process. As the absolute sensitivity (smallest single reflection/backscattering that can be detected) is usually the parameter of most interest in OCT systems, it is standard to measure the noise level with no sample present. The modeled and measured spectra are then processed with the same A-Scan reconstruction process, and the temporal standard deviation of the pixel values gives the noise value.

Knowing the relative (to reference) sample amplitude at which (or just before), with constructive interference, detector pixels start saturating (in our case a 1:1 ratio), the expected signal can be modeled using the basic interference equation. Again, using the same A-Scan reconstruction procedure and adjusting the modeled interference signal to be at the same depth, the peak signal is compared between the experimental and modeled values.

Figure A1 provides the MATLAB function to model the raw shot noise for a given detector well depth and raw output. It demonstrates its use with a hypothetical spectral shape for an SD-OCT system.

For a hypothetical FD vs. TD SNR experiment, the hardest (most subjective) part is to define equivalent FD and TD setups. Here, we define that the A-Scan depth and number of raw measurement points are equal. Due to Nyquist limitations, this means that the resultant sampling density in the TD A-scan is twice that in the FD A-scan. The maximum number of detected photo-electrons for a raw (interference) data point was made to be the same; overall, this means that the total TD signal is a larger amount of light. The (arbitrarily selected, any could be inputted) spectral shape used for the simulation was Hann. FD A-scans were reconstructed using an FFT, while TD A-scans were reconstructed using a Hilbert transform. The simulations were run with N values of 400 to 2000, in 200 increments. Figure A3 contains the full code of this simulation.

### 2.2. Lateral Resolution Modeling

Numerically modeling the expected image of a known object (target) with an imaging system, with an estimated PSF, is just a convolution operation. Figure 2 shows a flow chart of our modeling method. Here, we approximate the PSF as a Gaussian function, with different resolutions in the two dimensions. For the physical system used here for experimental comparison, in the line field dimension (i.e., resolution determined solely by imaging optics), the expected PSF would be an Airy function, which is well approximated by a Gaussian function. The confocal dimension (i.e., resolution determined by illumination gating (approximately Gaussian beam) and imaging) will also closely approximate a Gaussian. However, for other imaging systems, any arbitrary PSF shape could be used. With a modeled perfect image of the USAF element and PSF (each with the same model element scaling), we used MATLAB's conv2 function to perform the convolution. This gives the modeled optical image; the next consideration is the finite pixel size. The image is

binned into the pixels; for the confocal dimension, here, the galvo mirror was continually scanned during the collection, but the duty cycle of the camera was only about 9%, so this was also considered in the binning. In the line field dimension, we assume a negligible boundary between pixels, i.e., 100% duty. For the confocal direction, the Nyquist sampling limit is the main limiting factor. For the three bar gratings, within approximately ±10% of the Nyquist limit, the positioning of the pixels relative to the grating bars (phase) can determine if the grating is resolved or not in the image. So, control of this phase is also included in the model.

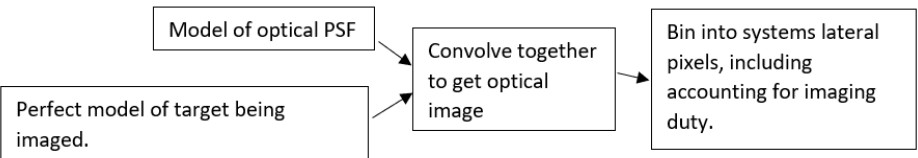

**Figure 2.** Flowchart of numerical convolution modeling of an expected target image for a given lateral PSF.

Figure A2 includes the MATLAB code for modeling the expected output image of a given USAF 1951 element, a Gaussian PSF of the inputted FWHM, and pixel size, duty, and phase to grating. This is 2D, with independent parameters that are settable in both dimensions. The example models presented in this paper are included as a demonstration. In our case (and likely most cases), where the orthogonal lateral resolutions act independently, simpler 1D convolutional operations can be used to give identical results to those of full 2D modeling. We use 1D modeling to show the effect of the optical resolution of a USAF 1951 element and two point-objects separated by its period as a Gaussian PSF FWHM is increased to greater than the period. Figure A4 provides the MATLAB code for this simulation.

### 2.3. Hardware Used for Validation

To demonstrate the use of these direct numerical models as a useful assessment tool during system development, we have used an arbitrary optical setup that is still in the early stages of development, which is what we had available at the time of writing this paper. It is at a point in development where assessment against direct numerical modeling highlights definitively where there are and are not issues in the construction so far.

In brief, Figure 3 gives a broad overview of the prototype LF-SD-OCM setup used. It is a compacted slit-less design, which means that there is interdependence (due to the shared final lens) between the performance of the microscope and the spectrograph. As a result, to achieve the prospective project's specification requirements, the design is a compromise. This includes that the full resolution potential of the microscope objectives used will not be used (ultimately due to Nyquist sampling limitations).

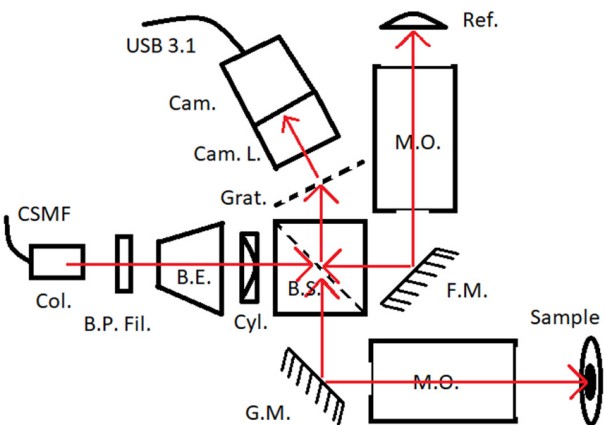

**Figure 3.** Schematic diagram of the LF-SD-OCM hardware in development used for demonstration here. CSMF—continuous single-mode fiber of a low-cost supercontinuum light source (Whitelase Micro, Fianium, Southampton, UK), Col.—free-space collimator output of the light source. B.P. Fil.—bandpass filter (750 to 850 nm). B.E.—beam expander (x5), Cyl.—cylindrical lens (f = 50 mm), B.S.—cube beam splitter (50:50), G.M.—single-axis galvo scanning mirror, F.M.—folding mirror, M.O.—microscope objectives (MY20X-824, Mitutoyo, Kanagawa, Japan), Ref.—glass reference surface, Grat.—grating (300 L/mm), Cam. L.—camera lens (f = 100 mm), Cam.—CMOS camera (Alvium 1800 U-52m, Allied Vision, Stadtroda, Germany), USB 3.1—data interface for PC. Red arrows indicate optical paths.

## 3. Results

### 3.1. OCT SNR: Practical Evaluation

The sample for the system SNR evaluation was a glass interface, identical to the reference. The design integration time is chosen so that saturation occurs for surface reflections greater than this. Figure 4a,b show the measured interference signal, along with the calculated interference from the basic interference equation, after considering additional internal reflections that are present. It can be seen that there is an apparent coherence loss within the system. Although the apparent coherence is usually never perfect, this degree of signal loss is excessive and will need to be investigated during the system's development. The signal reduction is 37% (proportional to the electric field) (4.0 dB (power)).

Figure 4c,d show the measured and (shot) modeled noise values. The measured noise is 17% (1.4 dB) higher than the modeled noise, which is due to a combination of light-source RIN and camera read noise. This is inherent to the system design. Overall, the measured SNR (absolute sensitivity) is 68.5 dB (82.5 dB), while the numerical modeled shot noise limit is 73.8 (87.8) dB. Resolving the signal issue will bring these values to as close as 1.4 dB, which is the extra identified noise of the system.

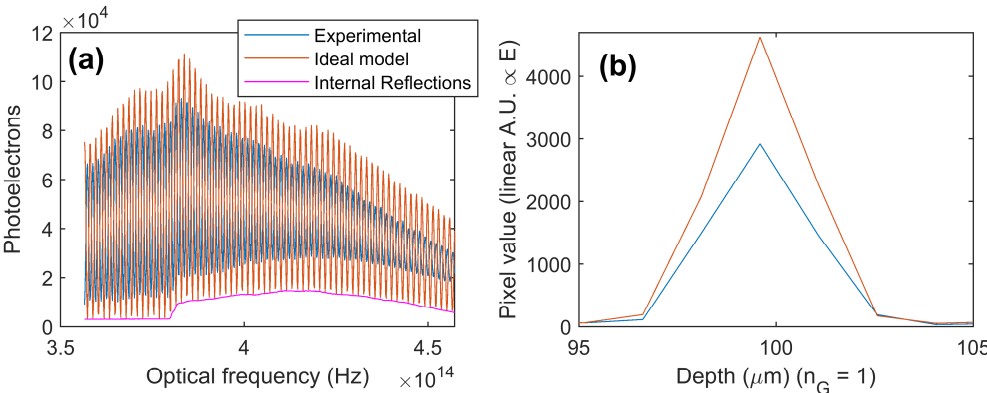

**Figure 4.** *Cont.*

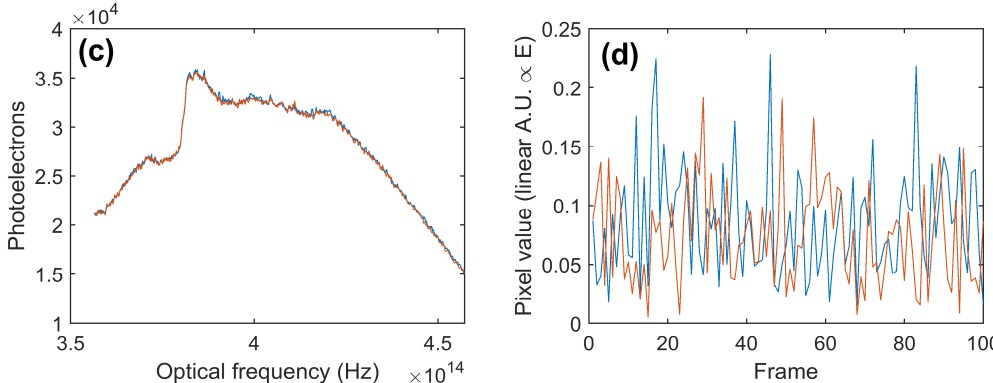

**Figure 4.** (**a**) Measured and predicted raw interference signal and (**b**) resultant signal amplitude difference after Fourier transformation. (**c**) An example of a measured and the corresponding directly numerically modeled raw reference spectra (used to quantify noise for absolute sensitivity) and (**d**) resultant (selected) pixel values over 100 frames.

### 3.2. OCT SNR: Theoretical Comparison of Time-Domain and Fourier-Domain Shot Noise Limits

Figure 5a shows the modeled raw signal for the FD and TD. Immediately, the reasoning for the performance difference is apparent. Where all measurement pixels for the FD system contain significant information, only three measurement pixels for the TD signal contain significant information. Figure 5b shows that the A-Scan signals are equivalent (arbitrarily displaced). Figure 5c shows the modeled SNR values; for the TD, the SNR is stable just above 50 dB independent of N. This makes sense, as the number of pixels carrying signal in the TD system for the hypothetical experiment does not change, so the extra data points do not give any extra information. For the hypothetical FD system, the SNR is dependent on N, as the extra pixels carry additional information. For N = 200, the improvement of the FD OCT over the TD is just less than 15 dB; for N = 2000, this increases to over 20 dB. Figure 5d shows how the difference factor between the (power) SNR of the FD and TD varies with N. A dependence on N is expected, with a linear fit giving it for the modeled conditions as 0.113 N.

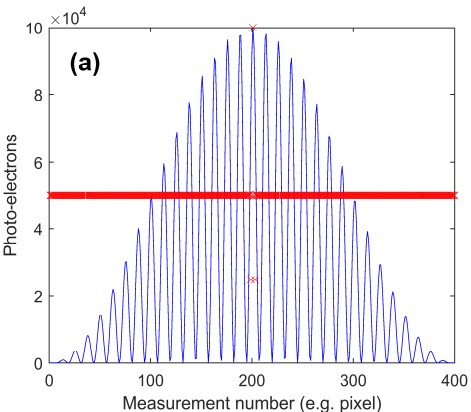
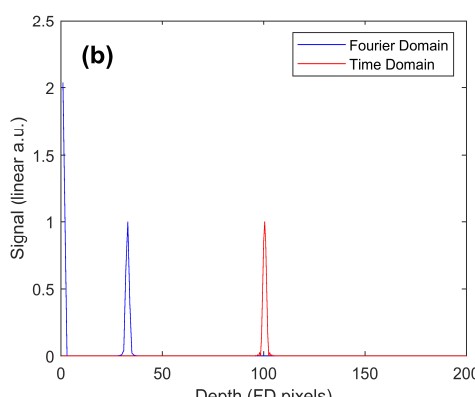

**Figure 5.** *Cont.*

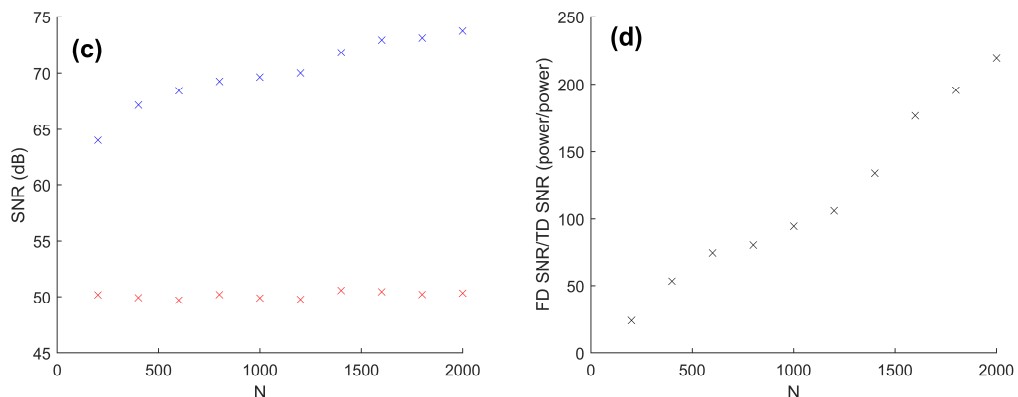

**Figure 5.** Direct numerical modeling of the shot noise SNR limit of Fourier domain vs. time domain OCT. (**a**) An example (N = 400) of the equivalent raw signal data for the (blue line) Fourier-domain and (red crosses) time-domain OCT A-Scans of the same scan depth, with the same number of raw data points and the same peak raw signal (the model time domain system collects more photo-electrons overall). (**b**) Both A-Scan signals after processing and normalization. (**c**) Modeled SNR (intensity) results vs. the number of measurement values (N) for both methods. (**d**) The signal-to-noise (power) factor benefit of FD OCT over TD OCT for the modeled conditions.

### 3.3. Lateral Resolution

Figure 6 shows the numerical resolution modeling of the finest vertical (line field) and horizontal (confocal) USAF elements that were visually resolved experimentally with the LF-SD-OCM system, with just better than the required resolution. For the line field dimension, element 8-5 was the smallest resolved in the image. The modeling shows that this is just resolvable (three discrete peaks) with an optical FWHM resolution of 2.4 μm and system-estimated pixel binning. The measured resolution of the peaks is clearer than the modeled resolution, so actual optical resolution is likely a little (but not too significantly) better. This is a little worse than the designed target resolution (2 μm) and may be indicative of an optical misalignment(s) within the system, possibly the same cause as that of the loss of apparent coherence.

In the confocal dimension, the limiting factor is sampling rather than optical resolution. In fact, for the resolution in 9-3 to be as good as they are in the experimental results, the B-Scan sampling must be denser than designed (so the overall image volume size is smaller than designed). However, when calibrated, to meet the design specifications, the sampling distance in the final device will be increased, decreasing the resolution. In this dimension, the resolution of the system is already sampling limited, so, optically, there are no concerns to address.

Figure 7a shows the 1D model version of Figure 6a, with three peaks that are visibly resolved. Figure 7b shows the model image of two point objects separated by the period, and two peaks are resolved. Increasing the PSF FWHM to just above the period value, both the three bars (Figure 7c) and point objects (Figure 7d) are still resolved. However, for an FWHM that is 20% greater than the period, both are no longer resolved (Figure 7e,f). The period of the smallest USAF 1951 element resolved is consistent with the distance at which two point objects can be resolved.

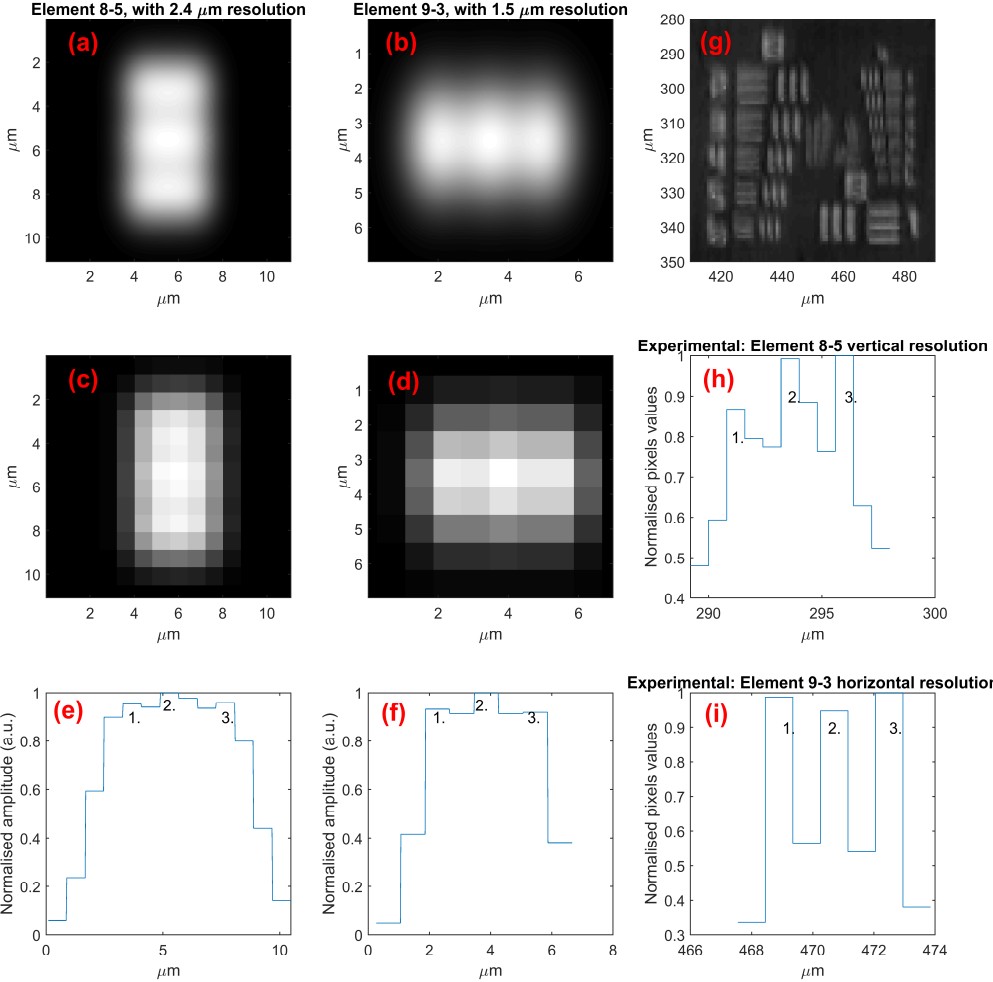

**Figure 6.** Direct numerical (convolution) modeling of (**a**,**b**) physical optical resolution and (**c**,**d**) after accounting for pixel binning for vertical (**a**,**c**) and horizontal (**b**,**d**) resolution elements, with the modeled resolution being just within the resolving limit. (**e**,**f**) Intensity plot across elements in (**c**,**d**) showing the three lines being resolved as just three peaks. (**g**) Experimentally measured (maximum projection surface) image of groups 8 and 9 of the USAF 1951 target, with (**h**,**i**) showing the actual measured intensity profile across the elements for comparison with the modeled values in (**e**,**f**). The numbers (1., 2., 3.) indicate the three resolved peaks in the images.

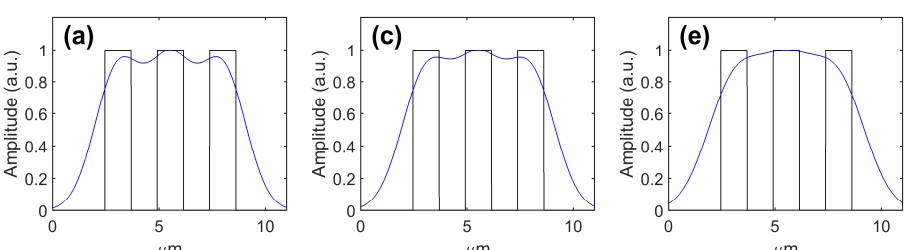

**Figure 7.** *Cont.*

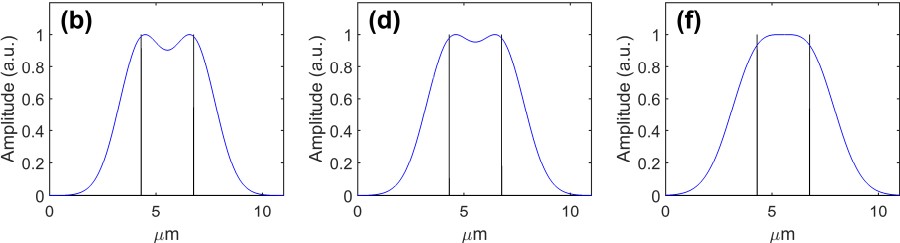

**Figure 7.** One-dimensional convolutional modeling of imaging (**a,c,e**) USAF 1951 element 8-5 (period of 2.46 μm) with a Gaussian PSF with an FWHM (**a,b**) less than (2.4 μm), (**c,d**) less than 20% greater than (2.55 μm), and (**e,f**) more than 20% greater than (2.9 μm) the period; and (**b,d,f**) equivalent point objects with a separation of 2.46 μm. Black lines show the original object. Blue lines show convolution images.

## 4. Discussion and Conclusions

Direct numerical modeling, starting from the detected signal, of the expected OCT SNR (and, thus, sensitivity) is a relatively simple and useful tool for system validation to help identify the presence of any issues. This decouples SNR considerations from optical efficiency through the device, which can (and should be) measured and dealt with separately.

The most cited OCT SNR papers, those of Leitgeb et al. [26] and Choma et al. [25], provide equations where the user plugs in the starting power and all the devices' transmission efficiencies. To give a direct comparison to this work for the shot noise SNR limit of SD-OCT, their solution (note: linear to power, not the electric field) turns out to be half of the total detected photo-electrons from the sample. However, for example, in the case of Figure A1, applying this underestimates the SNR by 10 dB. Similarly, Choma et al. [25] conclude that, for a Gaussian-like source, there should be a 0.25 N power SNR improvement for the SD over the TD. However, under our modeled experimental conditions, as shown in Figure 5d, we found a relationship of 0.11 N. This discrepancy is largely due to the larger overall total light energy in our model TD system compared to the FD system (we kept the peak point raw signal equal instead). Though not currently standard in the literature, using random number generation with a Poisson distribution to simulate shot noise is a relatively simple concept. Ossowiski et al. [32] and our group [33] previously used the technique in work on other topics. However, we are not aware of prior publications specifically evaluating and detailing the technique in the OCT field. It can be noted that, here, we did not model other noise sources, instead quantifying it as the difference between the experimental results and the modeled shot noise. Adding noise terms to the model directly for camera read noise and the light source requires more data and analysis to get accurate models of each term. Shot noise is the dominant noise source for the majority of OCT systems, and being at the shot noise limit is the target for OCT system developers [34]. Here, we provide a simple novel code base that can be quickly adapted and expanded to determine the theoretical SNR performance of any OCT system. Direct numerical modeling can be applied to any spectrum, and there are no assumptions about spectral shape.

For many conventional FD-OCT systems/applications, a performance metric of interest related to the SNR is the roll-off. This is a measure of how the signal (therefore, the SNR) is lost relative to the axial position in the image. For a real system in use, there are two main mechanisms; firstly, confocal or quasi-confocal gating means that light is lost and not meaningfully detected, and for OCM systems with optics with a low Depth of Field (DoF), this is significant. However, for conventional OCT with a high-DoF optical design, the high sensitivity of OCT (compared to the confocal effect) means that the second mechanism, finite spectrometer resolution, dominates. In a recent publication [33], we used a direct numerical model of convolving the expected signal with the measured (Gaussian fit) spectral PSF. The measured dB SNR curve was shown to correspond to the highest-resolution part of the spectrum (the spectral resolution was not uniform across the bandwidth), with the

side effect that the axial resolution also deteriorated with depth due to the effective loss of spectral bandwidth.

Numerical convolution modeling of a given imaging system's PSF gives the expected output image for the test target or modeled sample. Here, we used it to evidence the better interpretation of the USAF 1951 test targets. In OCT, where the desire is to present single resolution values, this method is sufficient to prove by modeling value interpretation. However, full characterization of the PSF/MTF remains the (scarcely used) gold standard for any imaging system. However, direct numerical modeling of a test target image using a measured PSF may still be used to verify this.

Though numerical convolution can be used just as easily for any arbitrarily shaped PSF, we note here that we used a Gaussian shape, as it was the best approximation for the experimental apparatus used for verification and, generally, to cover most other systems. For systems where the geometries of the optical components used are known and not proprietary, which generally is not the case for more complex objectives, including the ones used here, then the next possible step would be to model the geometrical and/or physical optical PSF from a device's optical design using software such as Zemax. Zemax, for example, has a built-in functionality to convolve its calculated PSF with a model target to get the estimated image. In the wider field of optics, further development of this concept is an active area of research [35,36]. However, such rigor does not yet exist in general in the OCT field, with examples in the peer-reviewed literature of disputable arbitrary interpretations of the USAF 1951 test target images. However, this is not a criticism of the USAF 1951 target itself, which is the most straightforward way of measuring unitary resolution values as long as it is used and interpreted in the correct way and has sufficiently high-resolution elements for the system under analysis. Historically, the lack of sufficiently high-resolution USAF 1951 targets used in publications has been an issue, but Newport now sells a high-resolution model (used here) that goes to a resolution beyond the reach of any reasonably conceivable optical system.

Section 3.1 provides a comparison of experimental data verifying that the shot noise model provides realistic outputs at all stages. The model itself is based on the basic and established physical principle of the Poisson noise of photons and has been used elsewhere with no issues identified. With the model established, Section 3.2 uses it to virtually perform an experiment on a given equivalence between TD and FD OCT, which would be impractical to do physically. The first half of Section 3.3 provides a comparison of experimental data showing that convolution modeling gives realistic outputs. Again, the principles of the model are well established. The second half of Section 3.3 uses the modeling method to provide precise answers on the theoretical resolution limit for the USAF 1951 target elements; it can resolve up to a 20% smaller period than the FWHM for a Gaussian PSF.

Overall, direct numerical modeling is a simple (with modern computational tools) and powerful tool for OCT system developers and researchers. Here, we have demonstrated it to assess how close a system's noise is to the shot noise limit, to predict the results of a hypothetical experimental setup to compare equivalent FD vs. TD systems' SNR performance, and to evidence how lateral resolution test target results should be interpreted. The models presented here should by no means be groundbreaking; however, in our experience of the OCT literature we have not encountered similar uses within the published literature (excluding our own work). Wider adoption would be part of the solution to provide a more robust analysis of present systems' performance against theoretical expectations.

**Author Contributions:** Conceptualization, S.L.; Methodology, S.L.; Software, S.L.; Writing—original draft, S.L.; Writing—review & editing, S.L. and Y.-C.S.; Supervision, Y.-C.S.; Project administration, Y.-C.S.; Funding acquisition, Y.-C.S. All authors have read and agreed to the published version of the manuscript.

**Funding:** The methodology presented has been developed and used within the following funded projects: NIHR II-LA-0813-20005 and II-LA-1116-20008, as well as EPSRC EP/X01441X/1 and EP/W006405/1.

**Institutional Review Board Statement:** Not applicable.

**Informed Consent Statement:** Not applicable.

**Data Availability Statement:** Model code is provided in Appendix A. If you require a copy of the example experimental data used within this paper, please send your request to the correspondence email address.

**Conflicts of Interest:** The authors declare no conflicts of interest.

## Appendix A

Figure A1 provides the MATLAB code for calculating the shot noise's absolute sensitivity limit for a given experimental setup. The parameters are set similarly to those used in the modeling in Section 3.1, but a Hann-shaped spectrum is used rather than the experimentally measured one, and additional internal reflection is not accounted for. Figure A2 provides the MATLAB code for the modeled parts of Figure 6. Figure A3 provides the code for producing Figure 5. Figure A4 provides the code for producing Figure 7.

```matlab
%% Detector specs
PixelElectronWellDepth = 100000;
ADC_BitDepth = 12;
%% Calculate conversion factor
conv_fac = PixelElectronWellDepth/(2^ADC_BitDepth);
%% Simulated reference spectra, in ADC units
A = 1000; % with equal sample intensity, this will give a maximum raw signal just below saturation
Ref_ADC = A.*hann(624); % arbitrary convenient spectral shape
%% Convert to photoelectrons
Ref = conv_fac .* Ref_ADC;
%% Create signal (for simplicity here we will assume we are already in equal K/frequency space)
As = 1; % relative (to reference) sample arm amplitude (energy)
Samp = As.*Ref;
QuasiDepth = 0.2;
Signal_raw = Ref+Samp+2.*sqrt(Ref).*sqrt(Samp).*cos(QuasiDepth.*(1:length(Ref)))'; % basic interference
equation
Signal_OCT = abs(fft(Signal_raw)); % model OCT signal
[Sig,loc] = max(Signal_OCT(5:end/2)); % measure peak signal and location
loc = loc+4;
%% Model noise (no sample) at same location
Noise_data = zeros(1000,1);
for N=1:1000
    temp = poissrnd(Ref);
    temp = abs(fft(temp));
    Noise_data(N)=temp(loc);
end
Noise = std(Noise_data);
%%
SNR_liner = Sig/Noise; % proportional to electric field
SNR_dB = 20*log10(SNR_liner); % relative to power
SamplePowerReflectionCoefficient = 0.04;
AbsoluteSensitivity_dB = SNR_dB - 10*log10(SamplePowerReflectionCoefficient)
```

**Figure A1.** MATLAB code example for a simplified ideal shot SNR (sensitivity) model.

```
close all
clear
%% example vertical resolution element 8-5
% call the model code, see function below.
[out,ax_xy,out_pix]=LatRes_Sim(8,5 ...
    ,1.5,0.8,0.000280/(1/(488/1.5)),0 ...
    ,2.40,0.8,1,0.5);
% display of the output in figure 1
figure(1);
colormap gray
% display numerically modelled optical image
subplot(3,2,1)
imagesc(ax_xy,ax_xy,out)
xlabel('\mum')
ylabel('\mum')
title('Element 8-5, with 2.4 \mum resolution')
% display after binning into finite pixel size and duty.
subplot(3,2,3)
imagesc(ax_xy,ax_xy,out_pix)
xlabel('\mum')
ylabel('\mum')
% intensity profile across the elements
subplot(3,2,5)
plot(ax_xy,out_pix(:,end/2)./max(out_pix(:,end/2)))
xlabel('\mum')
ylabel('Normalised amplitude (a.u.)')
% three peaks (bars) are just resolved and countable
text(3.5,0.9,'1.')
text(5,0.95,'2.')
text(7.5,0.9,'3.')

%% example horizontal resolution element 9-3
% Note that the moddelling code only creates the element in one direction,
% so we need to rotate everything here around by 90 degrees.
% call the model code, see function below.
[out,ax_xy,out_pix]=LatRes_Sim(9,3 ...
    ,2.4,0.8,1,0 ...
    ,1.5,0.8,0.000280/(1/(488/1.5)),2.0);
% display numerically modelled optical image
figure(1)
subplot(3,2,2)
imagesc(ax_xy,ax_xy,rot90(out))
xlabel('\mum')
ylabel('\mum')
title('Element 9-3, with 1.5 \mum resolution')
% display after binning into finite pixel size and duty.
subplot(3,2,4)
imagesc(ax_xy,ax_xy,rot90(out_pix))
xlabel('\mum')
ylabel('\mum')
% intensity profile across the elements
subplot(3,2,6)
plot(ax_xy,out_pix(:,end/2)./max(out_pix(:,end/2)))
xlabel('\mum')
ylabel('Normalised amplitude (a.u.)')
% three peaks (bars) are just resolved and countable
text(2.1,0.9,'1.')
text(3.5,0.95,'2.')
text(5.2,0.9,'3.')
```

Continued on next page…

**Figure A2.** *Cont.*

```matlab
%% modelling code
function [out,ax_xy,out_pix] =
LatRes_Sim(group,element,x_sys_res_um,x_pixel_size_um,x_pixel_duty,x_pixel_phase...
    ,y_sys_res_um,y_pixel_size_um,y_pixel_duty,y_pixel_phase)
figure(10) % we're going to display process in seperate figures to make clear
% first, calculate target's period from group and element number.
period = 1e3./(2^(group+(element-1)/6)); % in um
% more parameters defined and calculated
model_samps_per_period = 100; % defines granularity of model
sampling_res = period./model_samps_per_period; % converts to um
width = 9*model_samps_per_period./2; % width (in elements) of model image
height = width; % height is the same
ax_xy = sampling_res .* (1:width); % both image axes scalling in um
% compute x dimension PSF from input
x_sys_res = x_sys_res_um./sampling_res; %convert to sampling points
x_stdy = x_sys_res./(2*sqrt(2*log(2)));
% std = (N-1)./(2*alpha);
x_alpha = (width-1)./(2*x_stdy);
x_spot = gausswin(width,x_alpha);
% compute y dimension PSF from input
y_sys_res = y_sys_res_um./sampling_res; %convert to sampling points
y_stdy = y_sys_res./(2*sqrt(2*log(2)));
% std = (N-1)./(2*alpha);
y_alpha = (width-1)./(2*y_stdy);
y_spot = gausswin(width,y_alpha);
% combine into a two dimensional optical PSF.
spot = y_spot*x_spot';
% display 2D PSF
subplot(2,2,1)
imagesc(ax_xy,ax_xy,spot)
xlabel('\mum')
ylabel('\mum')
% create perfect target image
target = zeros(width,height);
target([1.0*model_samps_per_period:1.5*model_samps_per_period...
    2.0*model_samps_per_period:2.5*model_samps_per_period...
    3.0*model_samps_per_period:3.5*model_samps_per_period]...
    ,1.5*model_samps_per_period:3.0*model_samps_per_period)...
    =1;
% display perfect target image
subplot(2,2,2)
imagesc(ax_xy,ax_xy,target)
xlabel('\mum')
ylabel('\mum')
% compute convolution
out = conv2(target,spot,'same');
% display optical image
subplot(2,2,3)
imagesc(ax_xy,ax_xy,out)
xlabel('\mum')
ylabel('\mum')
% display intensity profile across target to see if 3 peaks (bars) are
% optically resolved
subplot(2,2,4)
plot(ax_xy,out(:,round(height/2))./max(out(:,round(height/2))))
xlabel('\mum')
% sanity plot to manually check FWHMs of PSF are what was inputted
figure(11)
subplot(1,2,1)
plot(ax_xy,spot(:,round(end/2))./max(spot(:,round(end/2))))
grid on
xlabel('\mum')
subplot(1,2,2)
plot(ax_xy,spot(round(end/2),:)./max(spot(round(end/2),:)))
grid on
xlabel('\mum')
% check if pixel size input variables have been included to process
if ~exist('x_pixel_size_um','var'), return; end
% calculations of pixel period, size (duty) and offset (phase) in x
x_pix_period_in_pixels = x_pixel_size_um/sampling_res;
x_pix_size_in_pixels = x_pixel_duty*x_pix_period_in_pixels;
x_off_set = (x_pixel_phase/(2*pi))*x_pix_period_in_pixels;
% calculations of pixel period, size (duty) and offset (phase) in y
y_pix_period_in_pixels = y_pixel_size_um/sampling_res;
y_pix_size_in_pixels = y_pixel_duty*y_pix_period_in_pixels;
y_off_set = (y_pixel_phase/(2*pi))*y_pix_period_in_pixels;
% create empty output
s = size(out);
out_pix = nan(s);
% do all the binning
for y = y_off_set:y_pix_period_in_pixels:(s(2)-y_pix_period_in_pixels)
    yrange_in = round(y+1:y+y_pix_size_in_pixels+1);
    yrange_out = round(y+1:y+y_pix_period_in_pixels+1);
    for x = x_off_set:x_pix_period_in_pixels:(s(1)-x_pix_period_in_pixels)
        xrange_in = round(x+1:x+x_pix_size_in_pixels+1);
        xrange_out = round(x+1:x+x_pix_period_in_pixels+1);
        out_pix(yrange_out,xrange_out)...
            = mean(mean(out(yrange_in,xrange_in)));
    end
end
% display binned output
figure(12)
imagesc(out_pix)
end
```

**Figure A2.** MATLAB code for the modeling of USAF1951 target images.

```matlab
WD = 100000;
reps = 500;
nn = 0;
for N=200:200:2000
    nn=nn+1;
    % FD signal sim
    Arm = (WD./4).*hann(N)';
    Sig = 2*Arm + 2*Arm.*cos(0.5*(1:N));
    FD_Sigs = abs(fft(Sig));
    [FD_Sig, loc] = max(FD_Sigs(10:end/2)); loc = loc+9;
    if N==400
    figure(10)
    subplot(2,2,1)
    hold off
    plot(Sig,'b')
    subplot(2,2,2)
    hold off
    plot(FD_Sigs(1:end/2)./FD_Sig,'b')
    end
    % FD shot noise
    noises = zeros(reps,1);
    for M=1:reps
        temp = poissrnd(Sig);
        temp = abs(fft(temp));
        noises(M)=temp(loc);
    end
    FD_Noise = std(noises);
    % TD signal sim
    df = 0.5/N;
    TD_Arm = ((WD./4)./sum(Arm)).*Arm;
    TD_Sigs = zeros(N+1,1);
    mm = 0;
    for f = -0.25:df:0.25
        mm = mm+1;
        tsig = 2*TD_Arm + 2*TD_Arm.*cos(f*2*pi*(1:N));
        TD_Sigs(mm) = sum(tsig);
    end
    TD_AScan = TD_Sigs - (WD./2);
    TD_AScan = abs(hilbert(TD_AScan));
    [TD_Sig, loc_TD] = max(TD_AScan);
    if N==400
    figure(10)
    subplot(2,2,1)
    hold on
    plot(TD_Sigs(1:end-1),'rx')
    xlabel('Measurement number (e.g. pixel)')
    ylabel('Photo-electrons')
    subplot(2,2,2)
    hold on
    plot(0.5:0.5:N/2,TD_AScan(1:end-1)./TD_Sig,'r')
    xlabel('Depth (FD pixels)')
    ylabel('Signal (linear a.u.)')
    legend('Fourier Domain','Time Domain')
    end
    % TD shot noise
    TD_Ref = (WD./4)+zeros(N,1);
    noises = zeros(reps,1);
    for M=1:reps
        temp = poissrnd(TD_Ref);
        temp = abs(hilbert(temp));
        noises(M)=temp(loc_TD);
    end
    TD_Noise = std(noises)';
    % SNR
    TD_SNR = (TD_Sig./TD_Noise); % proportional to E
    FD_SNR = (FD_Sig./FD_Noise); % proportional to E
    figure(10)
    subplot(2,2,3)
    hold on
    plot(N,20*log10(FD_SNR),'bx'...
        ,N,20*log10(TD_SNR),'rx')
    xlabel('N')
    ylabel('SNR (dB)')
    subplot(2,2,4)
    hold on
    plot(N,(FD_SNR./TD_SNR).^2,'kx')
    drawnow
    out(nn) = (FD_SNR./TD_SNR).^2;
    xlabel('N')
    ylabel('FD SNR/TD SNR (power/power)')
end
```

**Figure A3.** MATLAB code for the direct numerical modeling of the shot noise SNR limit of FD vs. TD OCT.

```matlab
close all
clear
group = 8; element = 5;
period = 1e3./(2^(group+(element-1)/6)); % in um
model_samps_per_period = 500; % defines granularity of model
sampling_res = period./model_samps_per_period; % converts to um
width = 9*model_samps_per_period ./2; % width (in elements) of model image
ax_xy = sampling_res .* (1:width);
% create perfect target images
target = zeros(width,1);
target([1.0*model_samps_per_period:1.5*model_samps_per_period...
    2.0*model_samps_per_period:2.5*model_samps_per_period...
    3.0*model_samps_per_period:3.5*model_samps_per_period])...
    =1;
target_point_objects = zeros(width,1);
target_point_objects([round(width/2 - model_samps_per_period/2)...
    round(width/2 + model_samps_per_period/2)]) = 1;
x_sys_res_um = 2.4;
x_sys_res = x_sys_res_um./sampling_res;
x_stdy = x_sys_res./(2*sqrt(2*log(2)));
x_alpha = (width-1)./(2*x_stdy);
x_spot = gausswin(width,x_alpha);
out = conv(target,x_spot,'same');
figure(1)
subplot(2,3,1)
plot(ax_xy,target,'k-'...
    ,ax_xy,out./max(out),'b-')
axis([0 11 0 1.2])
xlabel('\mum')
ylabel('Amplitude (a.u.)')
text(0.5,1.1,'(a)','FontSize',14,'FontWeight','bold')
subplot(2,3,4)
out = conv(target_point_objects,x_spot,'same');
plot(ax_xy,target_point_objects,'k-'...
    ,ax_xy,out./max(out),'b-')
axis([0 11 0 1.2])
xlabel('\mum')
ylabel('Amplitude (a.u.)')
text(0.5,1.1,'(b)','FontSize',14,'FontWeight','bold')
x_sys_res_um = 2.55;
x_sys_res = x_sys_res_um./sampling_res;
x_stdy = x_sys_res./(2*sqrt(2*log(2)));
x_alpha = (width-1)./(2*x_stdy);
x_spot = gausswin(width,x_alpha);
out = conv(target,x_spot,'same');
figure(1)
subplot(2,3,2)
plot(ax_xy,target,'k-'...
    ,ax_xy,out./max(out),'b-')
axis([0 11 0 1.2])
xlabel('\mum')
ylabel('Amplitude (a.u.)')
text(0.5,1.1,'(c)','FontSize',14,'FontWeight','bold')
subplot(2,3,5)
out = conv(target_point_objects,x_spot,'same');
plot(ax_xy,target_point_objects,'k-'...
    ,ax_xy,out./max(out),'b-')
axis([0 11 0 1.2])
xlabel('\mum')
ylabel('Amplitude (a.u.)')
text(0.5,1.1,'(d)','FontSize',14,'FontWeight','bold')
x_sys_res_um = 2.9;
x_sys_res = x_sys_res_um./sampling_res;
x_stdy = x_sys_res./(2*sqrt(2*log(2)));
x_alpha = (width-1)./(2*x_stdy);
x_spot = gausswin(width,x_alpha);
out = conv(target,x_spot,'same');
figure(1)
subplot(2,3,3)
plot(ax_xy,target,'k-'...
    ,ax_xy,out./max(out),'b-')
axis([0 11 0 1.2])
xlabel('\mum')
ylabel('Amplitude (a.u.)')
text(0.5,1.1,'(e)','FontSize',14,'FontWeight','bold')
subplot(2,3,6)
out = conv(target_point_objects,x_spot,'same');
plot(ax_xy,target_point_objects,'k-'...
    ,ax_xy,out./max(out),'b-')
axis([0 11 0 1.2])
xlabel('\mum')
ylabel('Amplitude (a.u.)')
text(0.5,1.1,'(f)','FontSize',14,'FontWeight','bold')
```

**Figure A4.** MATLAB code for the 1D modeling of the resolution for a Gaussian PSF.

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
