# Peer review of "Direct Numerical Modeling as a Tool for Optical Coherence Tomography Development: SNR (Sensitivity) and Lateral Resolution Test Target Interpretation"

_photonics, doi:10.3390/photonics11050419_

Round 1

Reviewer 1 Report

Comments and Suggestions for Authors

1) From a Reviewer's point of view, the authors have tried to make a standardization for Numerical modeling for OCT SNR and Lateral resolution calculation, which others can adopt to make a direct measurement. Although the method used by authors is somewhat novel, many researchers have proposed and demonstrated similar methods. It is good that the authors have made their work open by providing their code, which somewhat lacks originality and advancement. Still, overall, the topic in itself is advantageous for publication.

2) The authors have not mentioned or explained in detail about the SNR, lateral resolution, and other OCT parameter calculations put forth by others. 

3) SNR depth fall-off for the entire imaging depth achievable by the OCT system must be included. 

4) OCT measurement of biological samples' internal layer, which will serve as proof and efficiency of the proposed methodology, is not included.

5) Line 48: "comparative values, e.g. [23-26]." expand and explain these.

6) Lines 53-56: "Instead, here we show for two examples with modern computation tools, direct numerical modelling, based just on the basic/fundamental optical laws, can provide a direct theoretical expectation for an exact system", "and" is missing to clearly state the two examples of their modern computation tool used by authors

7) At line 60, Give the exact full form of SNR at least once before using the abbreviation.

8) lines 75-76: "the expected OCT SNR (and sensitivity) from", with the use of "and" the authors word it in such a way that SNR and sensitivity are two different parameters. The signal-to-noise ratio (SNR) is a physical measurement of the sensitivity of an imaging system. SNR is a comparison of a signal value in the presence of a signal with a value for system noise in the absence of a signal. It's best for the authors to be more careful in their phrasing. 

9) In Figure 3, the laser beam path is not shown

10) Given the limited novelty, authors are recommended to consider making this a short communication rather than a Full-length article.

Comments on the Quality of English Language

The use of English is okay, but the manuscript needs to be completely revised after extensive reading. The use of articles is missing in some places, use of unnecessary parenthesis (line 70), missing closing parenthesis (lines 61-62), and missing full forms first use of abbreviation, for instance, lines 60 (SNR) and 70 (LCI). 

Author Response

Please see attached response letter to editor and reviewers, along with redlined version of manuscript.

Reviewer 2 Report

Comments and Suggestions for Authors

This study provides a new tool and perspective for the theoretical performance evaluation of optical coherence tomography (OCT) equipment through direct numerical modeling methods.

Comment 1: This article proposes a simplified method for solving problems. Compare with existing technology to illustrate its advantages and possible limitations.

Comment 2: A new method was proposed to interpret the lateral PSF shape using the test chart. But the numerical model in this section seems to assume an idealized PSF shape. It is necessary to elaborate on the impact of hypotheses on the interpretation of experimental results.

Comment 3: Suggest adding a flowchart or providing more detailed step descriptions in the main text regarding the noise modeling process.

Comment 4: The specific configuration and material information description of the equipment are not detailed enough. Please add information about the light source, detector type, and optical component specifications.

Comment 5: Optical coherence tomography (OCT) combined with hyperspectral technology will be a promising research direction. As an emerging high-speed technology, hyperspectral sensors can provide richer sample spectral information, which helps to improve imaging speed and application diversity.For example, “A stare-down video-rate high-throughput hyperspectral imaging system and its applications in biological sample sensing”, it is suggested that hyperspectral systems should be discussed.

Comment 6: All references should be formatted to meet the journal's guidelines. Some references appear to be missing publication details (e.g., page number, issue number).

Comment 7: The appendix lacks a brief introduction to the functionality and purpose of the code.

Author Response

Please see attached reply letter to editor and reviewers, along with the redlined version of resubmitted manuscript.

Round 2

Reviewer 1 Report

Comments and Suggestions for Authors

All previously raised comments have been addressed.

Comments on the Quality of English Language

No major problems were detected in the use of the English language. 

Reviewer 2 Report

Comments and Suggestions for Authors

1. The paper focuses on optoelectronic noise, but excluding other noise sources such as random intensity noise (RIN) and electronic reading noise may lead to underestimation of actual noise values. It is recommended to consider incorporating these additional noise sources into the model for a more comprehensive evaluation of system performance.

2.The validation of numerical models relative to experimental results is crucial, but it appears to be limited in the provided examples.

3.Optical coherence tomography (OCT) combined with hyperspectral technology will be a promising research direction. A stare-down video-rate high-throughput hyperspectral imaging system and its applications in biological sample sensing, it is suggested systems should be discussed.

4.The assumptions of numerical models are not entirely detailed, especially regarding spectral and light source characteristics.
